# Profiling of RNAs from Human Islet-Derived Exosomes in a Model of Type 1 Diabetes

**DOI:** 10.3390/ijms20235903

**Published:** 2019-11-25

**Authors:** Preethi Krishnan, Farooq Syed, Nicole Jiyun Kang, Raghavendra G. Mirmira, Carmella Evans-Molina

**Affiliations:** 1Department of Medicine, Indiana University School of Medicine, Indianapolis, IN 46202, USA; prekrish@iu.edu; 2Herman B Wells Center for Pediatric Research, Indiana University School of Medicine, Indianapolis, IN 46202, USA; fsyed@iupui.edu (F.S.); lynn6251@gmail.com (N.J.K.); rmirmira@medicine.bsd.uchicago.edu (R.G.M.); 3Department of Pediatrics, Indiana University School of Medicine, Indianapolis, IN 46202, USA; 4Richard L. Roudebush VA Medical Center, Indianapolis, IN 46202, USA; 5Indiana University School of Medicine, 635 Barnhill Drive, MS 2031A, Indianapolis, IN 46202, USA

**Keywords:** islet-derived exosomes, mRNA, long noncoding RNA, small noncoding RNA, total RNA sequencing, small RNA sequencing

## Abstract

Type 1 diabetes (T1D) is characterized by the immune-mediated destruction of insulin-producing islet β cells. Biomarkers capable of identifying T1D risk and dissecting disease-related heterogeneity represent an unmet clinical need. Toward the goal of informing T1D biomarker strategies, we profiled coding and noncoding RNAs in human islet-derived exosomes and identified RNAs that were differentially expressed under proinflammatory cytokine stress conditions. Human pancreatic islets were obtained from cadaveric donors and treated with/without IL-1β and IFN-γ. Total RNA and small RNA sequencing were performed from islet-derived exosomes to identify mRNAs, long noncoding RNAs, and small noncoding RNAs. RNAs with a fold change ≥1.3 and a *p*-value <0.05 were considered as differentially expressed. mRNAs and miRNAs represented the most abundant long and small RNA species, respectively. Each of the RNA species showed altered expression patterns with cytokine treatment, and differentially expressed RNAs were predicted to be involved in insulin secretion, calcium signaling, necrosis, and apoptosis. Taken together, our data identify RNAs that are dysregulated under cytokine stress in human islet-derived exosomes, providing a comprehensive catalog of protein coding and noncoding RNAs that may serve as potential circulating biomarkers in T1D.

## 1. Introduction

Type 1 diabetes (T1D) is characterized by the immune-mediated destruction of the pancreatic β cells, resulting in the lifelong need for exogenous insulin treatment. Over the past 30 years, considerable effort has been directed toward identifying therapies capable of inducing T1D remission and immune tolerance. A handful of interventions have shown modest efficacy in preserving C-peptide secretion when initiated at stage 3 T1D onset [1,2,3,4]. Recently, an anti-CD3 monoclonal antibody (teplizumab) delayed the onset of T1D by approximately two years in high-risk individuals [5], providing support to the notion that T1D is a preventable disease. However, despite these recent successes, T1D clinical trial efforts continue to be challenged by marked heterogeneity in treatment responses among study populations. Similarly, there is significant heterogeneity in disease progression in autoantibody positive at-risk individuals, thus highlighting the need for improved methods to identify those who should be targeted with immunomodulatory interventions.

Biomarkers that reliably detect β cell stress and death may aid in dissecting T1D-related heterogeneity. In recent years, healthy as well as diseased cells have been shown to release a heterogenous population of membrane bound vesicles known as extracellular vesicles (EVs) [6]. EVs can be classified into several subtypes, including microvesicles, exosomes, and apoptotic bodies, based on their size, origin, biogenesis, and release pathways [7]. Exosomes, which are defined as an EV population with a diameter of ~100 nm, have been implicated in a variety of molecular pathways relevant to T1D, including immune regulation, antigen presentation, programmed cell death, and inflammation [7]. In addition, exosomes are known to transport cellular contents such as nucleic acids and proteins, which can be transferred to recipient cells where they can facilitate intercellular communication as well as disease pathogenesis. As the exosome cargo may also change during disease evolution, cell-specific exosome signatures may serve as circulating biomarkers [8].

In the context of type 1 diabetes, islet-derived EVs released during inflammatory β cell stress have been shown to contain β-cell-specific autoantigens, such as GAD65, IA-2, and proinsulin [9]. Exosomes may also act as a communication bridge between immune cells and insulin secreting cells. Recently, Guay et al. demonstrated that specific contents of exosomes released from lymphocytes can induce β cell apoptosis [10]. Likewise, pancreatic islets have also been shown to secrete EVs that act in an autocrine manner to regulate β cell proliferation and death [11].

Toward the goal of informing the development of exosome biomarkers in T1D, the objective of this study was to profile the various classes of protein coding and noncoding exosome-associated RNAs in human cadaveric islet-derived exosomes treated with IL-1β and IFN-γ, an ex vivo exposure model intended to mimic the proinflammatory milieu of T1D. Since EVs are found to be enriched in nucleic acid species, our primary aim was to profile mRNAs and noncoding RNAs, including long noncoding RNAs (lncRNAs), micro RNAs (miRNAs), piwi-interacting RNAs (piRNAs), transfer RNAs (tRNAs), and small nucleolar RNAs (snoRNAs). Our results show that islet-derived exosomes express multiple classes of RNAs, with mRNAs (and their associated fragments) being the most abundant class of long RNAs and miRNAs being the most abundant class among the known small noncoding RNAs. In addition to generating a RNA landscape of islet-derived exosomes in T1D, our results revealed alteration in the expression pattern of every class of RNA profiled, suggesting that newer classes of RNAs have the potential to emerge as circulating biomarkers for T1D.

## 2. Results

### 2.1. Characterization of Exosomes

The overall workflow for this study is shown in Figure 1. Exosomes were isolated from human islet cell culture supernatants using ExoQuick (SBI Biosciences, Palo Alto, CA, USA). The nanoparticle tracking analysis (NTA) of isolated exosomes showed a mean particle size of 131.56 ± 3.72 nm and 125.3 ± 6.01 nm from supernatants of human islets treated with or without cytokines, respectively (Figure 2a,b). Immunoblot analysis of isolated exosomes confirmed the presence of known exosomal markers tetraspanin CD9 and CD63 (Figure 2c). Electron microscopy analysis (TEM) was performed to visualize the exosomes and confirmed the presence of round double membrane vesicles measuring <150 nm in size in both cytokine and control-treated islets (Figure 2d).

### 2.2. Descriptive Statistics of Total RNA and Small RNA Sequencing Profiles

From the total RNAseq protocol, approximately 37 million and 34 million reads were detected from control and cytokine-treated samples, respectively. In both control and cytokine-treated human islet-derived exosomes, 69% of the reads mapped to protein coding mRNAs and ~21% of the reads mapped to long noncoding RNAs (Figure 3a). For both mRNAs and lncRNAs, the length distribution of the majority of the reads ranged from 72 to 76 nt in both control and cytokine-treated samples (Figure 3b). These 72–76 nt long RNAs represented fragments of full-length mRNAs and lncRNAs.

Similarly, on average, about 3.1 million reads and 3.8 million reads were aligned to human genome hg19 in control and cytokine-treated samples, respectively, from the small RNAseq protocol. When annotated to different classes of small RNAs (Figure 3c), ~20% of the reads annotated to miRNAs, followed by tRNAs (6.7%), piRNAs (0.8%), and snoRNAs (0.2%). The length distribution of reads mapping to miRNAs ranged between 12 and 24 nt, with the highest peak observed at 22 nt, indicating that the majority of reads mapped to mature miRNAs. However, reads mapping to piRNAs were distributed between 12 and 31 nt, the highest percentage of reads were for 30 nt long transcripts, conforming to the average read length of mature piRNAs (Figure 3d). Since snoRNAs and tRNAs are generally longer than miRNAs and piRNAs, the length of reads mapping to snoRNAs and tRNAs extended up to 90 nt and 77 nt, respectively (Figure 3e). The details of reads mapping to all samples from total RNAseq and small RNAseq protocols are provided in Appendix A, respectively.

### 2.3. Long and Small RNAs Are Differentially Expressed in Cytokine-Treated Islet-Derived Exosomes

Since the primary aim of our study was to identify potential circulating RNA biomarkers for T1D, we applied a stringent criterion to filter the reads mapping to different classes of RNAs. We retained only those RNAs that had at least one read count in all the samples profiled. By this criterion, we identified 17,013 mRNAs and 5711 lncRNAs from the total RNAseq protocol (Figure 4a), while 444 miRNAs, 175 piRNAs, 91 snoRNAs, and 167 tRNAs were identified from the small RNAseq protocol (Figure 4b). A total of 133 mRNAs, 31 lncRNAs, 19 miRNAs, 25 piRNAs, 8 snoRNAs, and 20 tRNAs were found to be differentially expressed with a fold change ≥1.3 (linear scale) and *p* < 0.05. The number of up- and down-regulated RNAs is reported in Figure 4c,d for long and small RNAs, respectively. In general, the proportion of up- and down-regulated mRNAs and lncRNAs were distributed fairly evenly. There were more upregulated miRNAs, while a higher proportion of piRNAs, snoRNAs, and tRNAs were downregulated. The fold change values and *p*-values for differentially expressed RNAs identified from the total RNAseq and small RNAseq protocols are provided in Appendix A, respectively.

### 2.4. Differentially Expressed RNAs Are Predicted to Play Key Functions Contributing to T1D Pathogenesis

#### 2.4.1. mRNAs

Among the top ten upregulated mRNAs were *ANG, CCDC107, USP39*, and *MAPK8IP1*, and among the top downregulated mRNAs were *RBP4*, *BTAF1*, *CCNI*, and *TXK*. Although not differentially expressed, we found IAPP and Ins, in addition to other β cell identity markers, such as *MAFA*, *NEUROD1*, *NKX6.1*, and *FOXO1*, and progenitor cell markers, such as *NEUROG3*, *PAX4*, and *SOX9*, to be present in islet-derived exosomes. The functions of differentially expressed mRNAs were predicted using Kyoto Encyclopedia of Genes and Genomes (KEGG) annotated pathways. We considered all the pathways identified. The top enriched pathways were “signaling pathways regulating pluripotency of stem cells”. This analysis also identified several critical pathways linked with diabetes pathophysiology, including calcium signaling, insulin secretion, insulin signaling, and TGF-β signaling. Representative pathways along with the genes involved in each pathway are shown in Table 1.

#### 2.4.2. lncRNAs

lncRNAs are known to play a role in a diverse set of functions, including regulation of gene expression and activation or inhibition of protein coding genes. Since there are no known predictive tools for the functional enrichment analysis of lncRNAs, one way to understand the function of lncRNAs is to identify the protein coding genes that correlate with the expression of lncRNAs. We used the variance stabilized transformed counts of differentially expressed mRNAs and lncRNAs to run a Spearman’s rank correlation and identified 362 pairs of lncRNA–mRNAs with r > 0.8 and *p* < 0.05 (Appendix A). Of these, 173 pairs showed a positive correlation, suggesting that if the expression of a lncRNA increases, the corresponding mRNA expression will also increase. However, 189 pairs were negatively correlated, indicating that these pairs shared an inverse relationship. Since differentially expressed mRNAs were used to predict the functions of lncRNAs, lncRNAs involved in the representative pathway are indicated alongside mRNAs in Table 1.

#### 2.4.3. miRNAs

Of the 444 miRNAs profiled in islet-derived exosomes, nineteen miRNAs were found to be differentially expressed. hsa-miR-155-5p, a well-studied inflammatory miRNA, was found to be the most upregulated miRNA, while hsa-miR-4485 was found to be the most downregulated miRNA. To define potential mRNAs and pathways regulated by these miRNAs, target predictions for differentially expressed miRNAs were performed using the miRNA target filter module in Ingenuity Pathway Analysis tool (IPA). In silico predictions were overlapped with the differentially expressed mRNAs to identify potential mRNA targets expressed within human islet-derived exosomes. A total of 66 mRNAs were identified as potential targets for differentially expressed miRNAs (Appendix A). Further, the core analysis module was used to predict the functions of miRNA–mRNA pairs. We focused on “diseases and functions” results and excluded terms related to cancer (e.g., neoplasia, cancer, neoplasm, and -oma). We found 19 functional terms with an activation z-score ≥0, and this included terms such as “inflammation”, “synthesis of proteins”, “necrosis”, “synthesis of nitric oxide”, “differentiation of stem cells” and “cytostasis”. The remaining predicted terms with no activation score were not considered further. The list of functional terms with the genes and the corresponding miRNAs are given in Appendix A.

#### 2.4.4. piRNAs

piRNAs are a class of small noncoding RNAs that associate with a class of Argonaute proteins called P-element Induced WImpy testis (PIWI) proteins for their biogenesis and function. piRNAs have predominantly been linked to germ cell/stem cell development and maintenance and transposon regulation [12,13,14,15]. However, several studies have also demonstrated their gene regulatory potential in somatic cells [16,17]. Furthermore, the stability of piRNAs in body fluids has also opened up the possibility that piRNAs might serve as circulating biomarkers [18,19].

mRNA targets were predicted using the miRanda algorithm, with the focus on inverse relationship targets, i.e., if a piRNA was upregulated, the corresponding mRNA targets were identified from the group of downregulated genes and vice versa. Functions of the 25 differentially expressed piRNAs were predicted using the same workflow used to predict miRNA targets. Similar to miRNAs, 16 functional terms (exclusive of terms related to cancer) with an activation score ≥0 were identified for piRNA targets (Appendix A). Since miRNAs and piRNAs share a similar gene regulatory mechanism by binding to the 3′UTR of mRNAs, we aimed to identify the common functions that could potentially be perturbed by both miRNAs and piRNAs. Eight functional terms were common between miRNA and piRNA targets, suggesting a possible co-regulation of the pathways by two classes of small noncoding RNAs. The common functional terms included “synthesis of proteins”, “quantity of mononuclear leukocytes”, “quantity of lymphocytes”, “differentiation of adipocytes”, “necrosis”, “cell survival”, “cell viability”, and “cellular homeostasis”. The representative common functions between miRNA and piRNA targets are illustrated in Figure 5a. Additionally, 14 and 13 functional terms were unique to miRNA and piRNA targets, respectively. The representative unique functions of miRNAs and piRNAs are illustrated in Figure 5b,c, respectively.

#### 2.4.5. snoRNAs and tRNAs

Since miRNAs and piRNAs have been demonstrated to be processed from larger RNAs such as snoRNAs and tRNAs [20,21,22,23], it is hypothesized that snoRNAs and tRNAs may play an indirect role in gene regulation. However, neither of these classes of small noncoding RNAs have been well studied in the β cell or in T1D pathogenesis. In order to characterize these mechanisms in the context of T1D, we overlapped the genomic boundaries of differentially expressed snoRNAs and tRNAs with the genomic boundaries of miRNAs and piRNAs. One snoRNA (HBII-436: piR-36717 (DQ598651)) and three tRNAs (tRNA-Lys-TTT-2-1: piR-35463(DQ597397); tRNA-iMet-CAT-2-1: piR-35176(DQ597110); and tRNA-Gly-GCC-2-4: piR-61648(DQ595536)) were identified to harbor piRNAs. While HBII-436 and tRNA-Lys-TTT-2-1 were found to be downregulated along with the embedded piRNAs, tRNA-iMet-CAT-2-1 and tRNA-Gly-GCC-2-4 were found to be upregulated along with the embedded piRNAs. Interestingly, none of the differentially expressed miRNAs were predicted to arise from within the genomic boundaries of snoRNAs or tRNAs.

## 3. Discussion

Reliable biomarkers capable of dissecting T1D-associated heterogeneity, guiding immunomodulatory interventions, and identifying individuals at the highest risk of disease remains an unmet clinical need. The advent of sequencing technologies has made it possible to obtain an overview of multiple classes of RNAs within biological samples, spurring interest in the potential for protein coding and noncoding RNAs to serve as noninvasive biomarkers of T1D (reviewed in [24] and [25]). Circulating RNAs, such as miRNAs, facilitate cell-to-cell communication and exert important biological functions via uptake of miRNAs by recipient cells, where they can act to modulate recipient cell gene regulatory function. In addition, the presence of different RNA species in the blood has been proposed to indicate cell lysis or turnover. Generally, RNAs are considered to be highly unstable as they are subjected to degradation by RNases. However, the encapsulation of nucleic acids in membrane bound vesicles, such as extracellular vesicles (EVs), limits the exposure of the encapsulated RNAs to degrading enzymes, thus making them more stable in blood and other bodily fluids. Indeed, earlier studies have demonstrated the release of EVs harboring nucleic acids from β cells undergoing stress [26]. In the case of T1D, specific nucleic acid signatures have been proposed to reflect processes such as β cell death and stress and serve as a proxy of residual β cell mass and C-peptide production [27,28,29,30]. Thus, the possibility of combining cell- and disease-specific nucleic acid signatures with technology that allows for the targeted identification of β cell-derived exosomes could have tremendous utility in T1D biomarker strategies.

Based on this premise, the goal of our study was to provide the first comprehensive map of mRNAs and noncoding RNAs (long and small) from human islet-derived exosomes. The study of T1D poses a unique challenge owing to the relative inaccessibility of the target organ for interrogation in living individuals. However, during the development of T1D, the local pancreatic microenvironment is enriched with invading immune cells, which are primed to release a number of proinflammatory cytokines and chemokines [31,32,33]. Thus, we utilized a well-accepted ex vivo stress model to mimic the proinflammatory milieu of T1D. This model involved treatment of human islets from nondiabetic cadaveric donors with the cytokines IL-1β and IFN-γ, which was followed by RNA sequencing to identify differentially expressed exosomal mRNAs, lncRNAs, miRNAs, piRNAs, snoRNAs, and tRNAs.

We identified this core set of RNAs by employing a stringent filtering cut-off of raw sequencing reads, and our objective was to identify potential candidates with expression in all the samples used for the study, increasing the likelihood of reproducibility. By this criterion, more than 50% of reads mapped to protein coding mRNAs. In fact, mRNAs were identified as the most abundant class of RNAs in islet-derived exosomes. However, our results showed that mRNAs are predominantly packaged as fragments in EVs. Earlier studies have shown that these fragments may be functional byproducts of full-length mRNAs, which upon entering the recipient cells may translate and become functional in this new location [34]. Therefore, the identification of mRNA fragments may still suffice the need to develop cell-specific signatures and to understand potential changes that cells undergo during EV release.

Total RNA sequencing also revealed the presence of lncRNAs in islet-derived exosomes. lncRNAs are an important class of noncoding RNAs, which play gene regulatory functions at the transcriptional, epigenetic, and post-transcriptional levels. However, the roles of lncRNAs in the context of diabetes remain less explored. Recently, Ruan et al. reported higher levels of lncRNA-p3134 in the blood of individuals with type 2 diabetes, when compared to nondiabetic controls [35]. The higher expression of lncRNA-p3134 also showed a positive correlation with fasting blood glucose, C-peptide, and HOMA-IR levels, indicating a strong link between diabetes development and the levels of circulating lncRNAs. Additionally, lncRNA-p3134 was also found to be associated with insulin secretion, β cell apoptosis, and glucotoxicity. In the context of type 1 diabetes, this is the first study to report a comprehensive list of lncRNAs in exosomes isolated from a model of T1D. Functional predictions of the differentially expressed lncRNAs based on the correlated mRNAs suggest roles in insulin secretion and other signaling pathways, such as the TGF-β signaling pathway.

In contrast to mRNAs and long noncoding RNAs, small noncoding RNAs, especially miRNAs and piRNAs, are more stable in circulation, due to their small size and their encapsulation in membrane bound vesicles [36,37]. In fact, several studies have reported the stability of miRNAs in serum samples stored for up to 40 years [38]. Among the four classes of small noncoding RNAs, ~21% of reads mapped to miRNAs. Being a discovery study, we used a filter of ≥1.3 fold change and *p* < 0.05 for identifying differentially expressed RNAs between untreated and cytokine-treated islet-derived exosomes. Among the differentially expressed RNAs, approximately 50% of miRNAs showed comparable results with those reported in literature. The top hit miRNA—miR-155-5p—has been found to be upregulated in blood from children with T1D who were aged 6–11 years, with a disease duration of 3.4 ± 1.9 years [39]. Similarly, miR-148a showed an elevated expression in serum and plasma of children with recent onset T1D, adults with long-duration T1D and individuals with latent autoimmune diabetes as well as those with pre- type 2 diabetes and type 2 diabetes (T2D). Further, miR-146a-5p [40,41], miR-802 [42], and miR-17-5p [43] were found to be upregulated in adults with T2D, similar to the expression pattern found in our analysis. Interestingly, some of the miRNAs identified in this study, such as miR-130b [44] and miR-223-3p [45], have been reported to be associated with diabetic complications, such as diabetic nephropathy or retinopathy.

Our data provide a comprehensive catalog of protein coding and noncoding RNAs in human islet-derived exosomes from an ex vivo model of diabetes. Previous studies have shown that the distribution of RNAs in exosomes is related to the parent cell type [46]. Since pancreatic islets are challenging to monitor for early disease development and progression, identification of molecular markers released from the islets may serve as indicators of a change in the cellular status. Circulating markers identified from patient-derived samples are often challenged with the question of identifying the source of these markers, because serum and plasma host a milieu of molecules from a variety of organs [47]. Studies such as ours may help narrow down the source of these circulating biomarkers and provide information about the trajectory of disease progression along with understanding the molecular functions of the identified biomarkers. However, the success of this approach will likely be dependent on the parallel ability to identify EVs that are β-cell-derived based on unique cell surface markers, as has been done in cancer [48]. Quite often, there has been some controversy on the choice of endogenous reference controls for validation studies in human serum or plasma samples [49]. Comprehensive profiling of RNAs from islet-derived exosomes may give us an overview of the expression patterns of RNAs and provide a good base to search for endogenous controls in biomarker studies related to T1D. While many studies have focused on miRNAs extensively, our study includes the identification of newer classes of potential biomarkers for T1D as well, such as piRNAs, snoRNAs, tRNAs, and lncRNAs, whose roles as biomarkers have been demonstrated in other disease settings [16,22,50,51,52,53,54]. Validation of the study findings in clinical samples is warranted as it may open up opportunities to explore beyond the well-studied classes of RNAs which can provide a basis for building a comprehensive and intermolecular biomarker model for T1D.

## 4. Materials and Methods

### 4.1. Isolation of Exosomes from Human Islets

To profile protein coding and long noncoding RNAs, human islets from five cadaveric donors were obtained from the Integrated Islet Distribution Program (IIDP), and their derived exosomes were processed for total RNA sequencing. Human islets from ten donors were obtained for exosome-derived small RNA sequencing; one islet donor was common between the two sequencing protocols. Islet donor and isolation characteristics, including donor age, sex, Body Mass Index (BMI), cause of death, measurements of islet purity and viability, ischemia duration, and culture time, are provided in Appendix A, as per recent recommendations for the reporting of human islet characteristics in research articles [55]. Human islets were provided from the IIDP in a de-identified manner. The study was reviewed and approved as exempt by the Indiana University School of Medicine Institutional Review Board. Upon receipt, cadaveric islets were cultured overnight in standard Prodo medium (Prodo Labs, Aliso Viejo, CA, USA) that had been depleted for exosomes. In order to mimic the proinflammatory conditions of T1D, islets were treated with or without 50 U/mL IL-1β and 1000 U/mL IFN-γ for 24 h. After 24 h of cytokine treatment, the islet culture media was collected and centrifuged for 30 min at 3000× *g* to remove cellular debris. The supernatant was recovered, and exosomes were isolated using ExoQuick-TC (SBI Biosciences, Palo Alto, CA, USA), according to the manufacturer’s protocol. The overall study workflow is illustrated in Figure 1.

### 4.2. Characterization of Exosomes

#### 4.2.1. Nanoparticle Tracking Analysis (NTA)

The concentration and size distribution of the islet-derived exosomes were measured using nanoparticle tracking analysis (ParticleMetrix, Inning am Ammersee, Germany) and analyzed using Zetaview Analysis software. Briefly, human islet-derived exosomes were isolated using the methods described above and maintained at −80 °C until batched analysis. On the day of analysis, samples were thawed over ice and diluted in distilled water (1:3000). One milliliter of diluted sample was injected for the measurement of exosome size and concentration using Zetaview software.

#### 4.2.2. Western Blot Analysis

Exosomes were isolated from human islet cell culture supernatant and lysed with protein lysis buffer. The protein contents were measured using the Lowry method. A total of 10 μL of protein per sample was electrophoresed on a 4–20% Bis-Tris gel (Bio-Rad, Hercules, CA, USA) under denaturing conditions and blotted onto a PVDF membrane. The blots were blocked using LICOR blocking buffer and probed for exosomal markers using the following primary antibodies: CD9 (SC5275, Santa Cruz, CA, USA, mouse monoclonal) and CD63 (SC13118, Santa Cruz, CA, USA, mouse monoclonal). LI-COR antimouse (1:10,000) secondary antibodies were used for the quantification of protein expression. The data were quantified using ImageStudio (LI-COR, Lincoln, NE, USA).

#### 4.2.3. Transmission Electron Microscopy (TEM) Analysis

Isolated exosomes were fixed with Karnovsky’s fixative (Electron Microscopy Sciences) and processed with a series of ethanol washes. Washed samples were exchanged with fresh fixative, microwaved, and then vacuum dried. Samples were washed again with cacodylate buffer and processed with a solution containing 1% OsO4, 0.8% KFe(CN)_6_. Next, the samples were washed twice with cacodylate buffer, followed by a wash with distilled water, then transferred into a solution of 1% UA in distilled water, and dehydrated with a series of ethanol washes. Finally, the samples were transferred into acetone, embedded in EMBed812 resin, and polymerized at 60 °C for 24 h. Ultrathin sections (90 nm) were collected on copper grids and imaged in a JEOL 1230 TEM with 80 kV accelerating voltage using a Gatan Orius camera.

### 4.3. RNA Isolation and Library Preparation

Total RNA was isolated using the SeraMir RNA isolation kit (SBI Biosciences, Palo Alto, CA, USA). RNA quantification and quality were assessed by measuring the 260/280 ratio using a nanophotometer (Implen, Munich, Germany). RNA integrity was evaluated using a Bioanalyzer 2100 (Agilent Technologies, Santa Clara, CA, USA). Total RNA sequencing utilized 1–10 ng of RNA as the input. Libraries for total RNA sequencing were prepared using Clontech SMARTer RNA Pico Kit v2, 96 rxns (Takara, Kusatsu City, Japan). Sequencing was performed using Illumina HiSeq 4000 (75 bp paired-end reads), generating 30 million reads per library.

For small RNA sequencing, 10–20 ng of RNA was used, and small RNA libraries were generated using the Ion Total RNA-Seq Kit v2 User Guide, Pub. No. 4476286 Rev. E (Life Technologies, Carlsbad, CA, USA). Each resulting barcoded library was quantified and then assessed for quality by an Agilent Bioanalyzer. Multiple libraries were pooled in equal molarity and 8 µL of 100 pM pooled libraries were applied to Ion Sphere Particles (ISPs) template preparation and amplification using Ion OneTouch 2, followed by ISP loading onto a PI chip and sequencing on an Ion Proton semiconductor. Sequence mapping was performed using Torrent Suite Software v4.6 (TSS 4.6).

### 4.4. Sequencing Data Analysis

Sequencing reads from both sequencing protocols were aligned to the human genome hg19 using STAR aligner v2.5.3a. Gencode v29 was used for annotating the total RNAseq reads to mRNAs and long noncoding RNAs. The following databases were used for annotating the different classes of small RNAs: mature miRNAs—miRBase v 20; piRNAs—National Center for Biotechnology Information (NCBI); snoRNAs—The University of California Santa Cruz (UCSC) genome browser; and tRNAs—tRNAdb. The annotation files for piRNAs, snoRNAs, and tRNAs were obtained from the integrated database—Database of small human noncoding RNAs: DashR v2.0 [56]. For all the RNA classes, only RNAs with a read count of at least one in all the samples were considered for downstream analysis. Partek^®^ Flow^®^ software version 7.0.18.1030 (Copyright ©; 2018 Partek Inc., St. Louis, MO, USA) was used for aligning reads to the reference genome, annotating the reads to respective databases, and for filtering reads for downstream analysis.

Differential expression analysis between untreated and cytokine-treated exosomes was performed for all RNAs using DESeq2 [57] paired sample analysis in the R statistical tool. Since this was a discovery study with the objective of profiling different classes of RNAs, we considered RNAs exhibiting a fold change of ≥1.3 (linear scale) and *p* < 0.05 as differentially expressed. The raw data files (fastq files), raw counts, and variance stabilized normalized counts for total RNA sequencing were deposited in the Gene Expression Omnibus (GEO accession ID: GSE139932). For small RNA sequencing, the raw counts and variance stabilized normalized counts are provided in Appendix A, respectively.

### 4.5. Gene Enrichment Analysis

Pathways for differentially expressed mRNAs were identified using KEGG pathways built within the Database for Annotation, Visualization, and Integrated Discovery (DAVID, http://david.abcc.ncifcrf.gov/) [58,59]. Differentially expressed lncRNAs were correlated with differentially expressed mRNAs. Spearman’s rank correlation was used for the analysis (“Hmisc” package in R statistical program), and lncRNA–mRNA pairs showing correlation values >0.8 and *p* < 0.05 were considered as significantly correlated pairs. mRNAs identified in the significant pairs were considered for pathway analysis using DAVID to understand the functions of lncRNAs.

Targets for differentially expressed miRNAs were predicted using the Ingenuity Pathway Analysis tool (QIAGEN Inc., Hilden, Germany, https://www.qiagenbioinformatics.com/products/ingenuitypathway-analysis). The in silico predicted targets were further overlapped with the differentially expressed mRNAs from the exosomes.

The targets of piRNAs were predicted using the miRanda algorithm. Complementary sequences between differentially expressed piRNAs and the 3′UTR of differentially expressed mRNAs were identified. piRNA–mRNA pairs with alignment scores ≥120 and energy scores ≤−15 kcal/mol (minus 15 kcal/mol) were considered for downstream analysis. The functions of miRNA–mRNA pairs and piRNA–mRNA pairs were predicted using the “diseases and functions” option within the IPA core analysis. For both miRNA and piRNAs, functional terms having any z-score were considered.

The function of snoRNAs and tRNAs were predicted based on the RNAs embedded within snoRNAs and tRNAs. The genomic boundaries of differentially expressed snoRNAs and tRNAs (host RNAs) were overlapped with the genomic boundaries of differentially expressed miRNAs and piRNAs (embedded RNAs). Further, the direction of expression between the embedded RNAs and the host RNAs were compared. RNA pairs showing the same direction of expression were considered for further analysis. We followed the same method for the functional analysis of the embedded RNAs as that of the previous functional analysis using miRNAs and piRNAs.

## Figures and Tables

**Figure 1 ijms-20-05903-f001:**
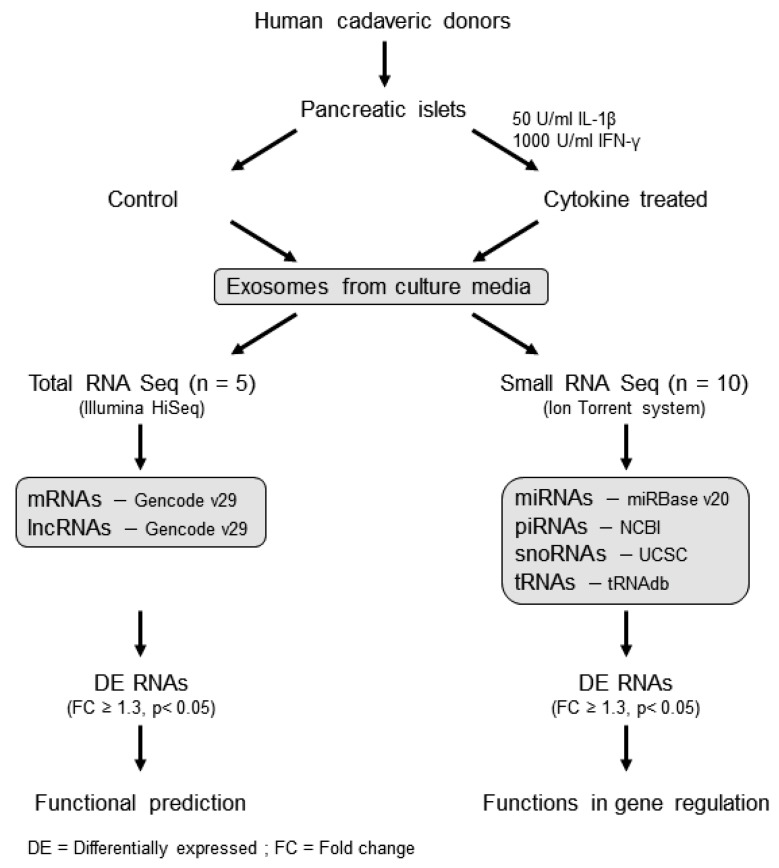
Workflow of the study: human islet-derived exosomes were treated with/without IL-1β and IFN-γ for 24 h and then subjected to total RNA sequencing and small RNA sequencing to profile protein coding and noncoding RNAs (long and small). The functions of differentially expressed RNAs were identified using the Ingenuity Pathway analysis tool.

**Figure 2 ijms-20-05903-f002:**
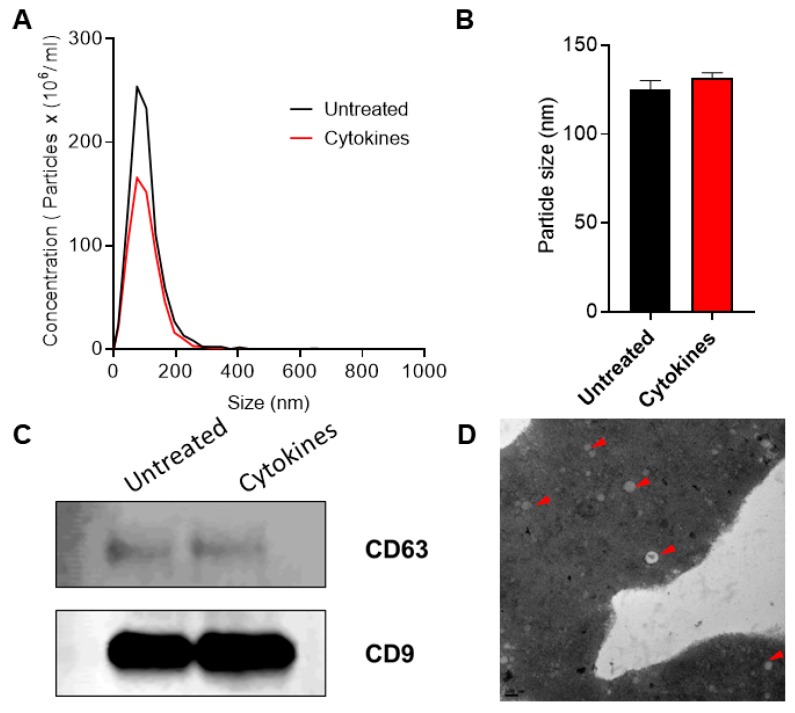
Characterization of exosomes: nanoparticle tracking analysis was performed to profile extracellular vesicle (EV) particle concentration (**A**) and size distribution (**B**) in control and human cytokine-treated islet-derived exosomes (*n* = 3), (**C**) immunoblot was performed using antibodies against CD63 and CD9, and (**D**) TEM imaging was performed to confirm the presence of exosomes. Shown is a representative image of exosomes from untreated islets. Representative exosomes are indicated with a red arrow. Scale bar: 100 nm.

**Figure 3 ijms-20-05903-f003:**
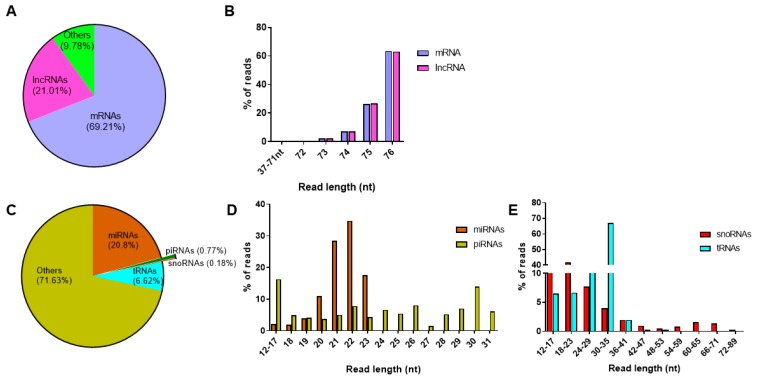
Profiling of long and small RNAs in exosomes: read distribution (**A**) and length distribution (**B**) of RNAs obtained from the total RNAseq protocol; read distribution of small RNAs (**C**) obtained from the small RNAseq protocol; and percentage of reads mapping to different read lengths are represented for microRNAs and piwi-interacting RNAs (**D**) and small nucleolar RNAs and transfer RNAs (**E**).

**Figure 4 ijms-20-05903-f004:**
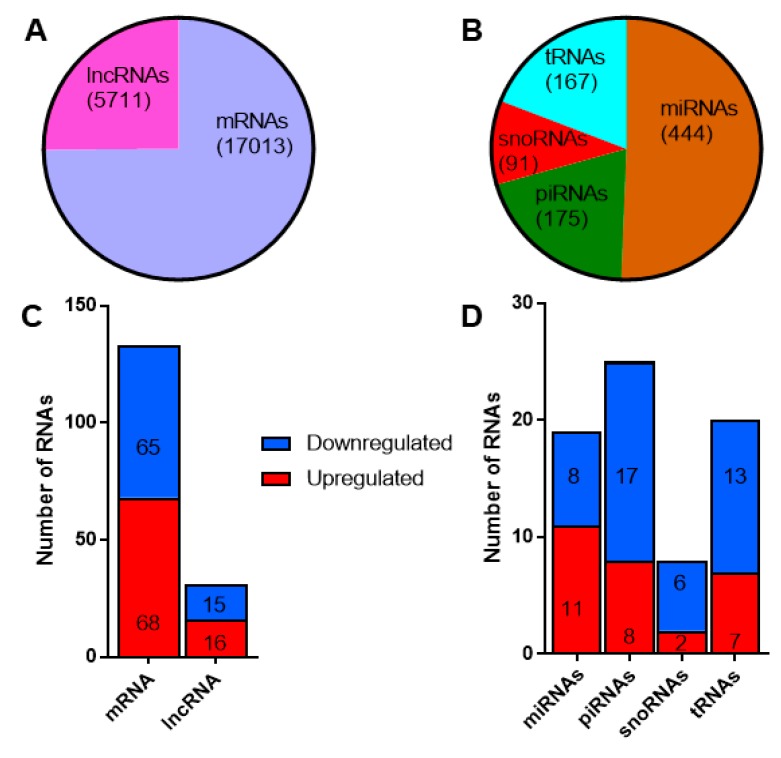
Differentially expressed long and small RNAs in exosomes: (**A**) the total number of long RNAs (mRNAs and long non-coding RNAs) and (**B**) different classes of small noncoding RNAs identified in exosomes. The number of differentially expressed (fold change ≥1.3, *p* < 0.05) long (**C**) and small RNAs (**D**). Shown in red are the number of upregulated long and small RNAs, while downregulated RNAs are shown in blue.

**Figure 5 ijms-20-05903-f005:**
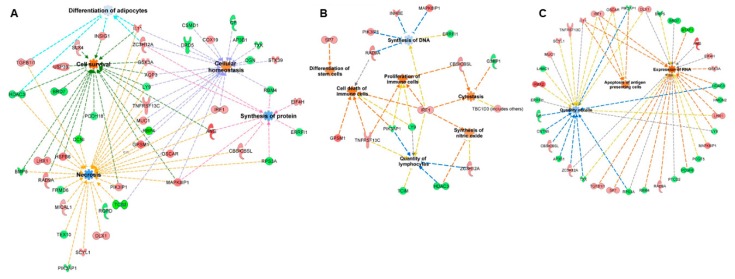
Functional enrichment analysis of miRNAs and piRNAs: miRNAs and piRNAs are considered master regulators of gene expression. Since the post-transcriptional mechanisms are similar between miRNAs and piRNAs, we identified the functions and target genes common to both classes of small noncoding RNAs (**A**) as well as functions unique to miRNAs (**B**) and piRNAs (**C**).

**Table 1 ijms-20-05903-t001:** Pathways identified for differentially expressed mRNAs and lncRNAs.

Pathway	Genes	lncRNA
Pancreatic secretion, Insulin secretion, Regulation of lipolysis in adipocytes	*ADCY8*	AC034229.2,AP001372.2,ITGB2.AS1
cAMP signaling pathway	*ADCY8*	AC034229.2,AP001372.2,ITGB2.AS1
	*DRD5*	
Chemokine signaling pathway	*ADCY8*	AC034229.2,AP001372.2,ITGB2.AS1
	*GSK3A*	AC089998.1,AL031123.1,LINC01783
Calcium signaling pathway	*ADCY8*	AC034229.2,AP001372.2,ITGB2.AS1
	*PHKG2*	AC239798.2,AL118505.1
	*DRD5*	
Lysosome	*AP3B1*	AC110079.2,AL121603.2
Biosynthesis of amino acids	*CBS*	AC024614.4,AC239798.2,AL591623.1
Hippo signaling pathway	*FRMD6*	AC089998.1,AC105275.2,LINC01783
	*DLG3*	AC010327.4,AL391987.3,CSNK1G2.AS1,LINC00967,LINC01258
	*BMP5*	AC110079.2,AL121603.2
Glycine, serine, and threonine metabolism	*GRHPR*	AC092620.3,AL031123.1
	*CBS*	AC024614.4,AC239798.2,AL591623.1
Hematopoietic cell lineage, Jak-STAT signaling pathway	*IL11*	AC004949.1,AP001043.1
TGF-beta signaling pathway	*INHBE*	AC004949.1,AC034229.2,AC105275.2,AP001372.2
	*BMP5*	AC110079.2,AL121603.2
Cytokine–cytokine receptor interaction	*INHBE*	AC004949.1,AC034229.2,AC105275.2,AP001372.2
	*TNFRSF13C*	AC110079.2,AL121603.2
	*IL11*	AC004949.1,AP001043.1
Extracellular matrix-receptor interaction	*LAMC1*	
Signaling pathways regulating pluripotency of stem cells	*PCGF5*	AC239798.2,AL118505.1,MAP3K14.AS1,RP11.680G24.5
	*INHBE*	AC004949.1,AC034229.2,AC105275.2,AP001372.2
	*PCGF6*	AC034229.3,RMRP
Insulin signaling pathway, Glucagon signaling pathway	*PHKG2*	AC239798.2,AL118505.1
B cell receptor signaling pathway	*PIK3AP1*	AC004852.2,AC008669.1,AC110079.2,AL391987.3
PI3K-Akt signaling pathway	*PIK3AP1*	AC004852.2,AC008669.1,AC110079.2,AL391987.3
	*LAMC1*	
Mitogen-Activated Protein Kinase signaling pathway	*PPP5D1*	AC110079.2,AL121603.2,AL391261.2,AP001043.1
	*MAPK8IP1*	AC089998.1,AC105275.2,LINC01783
Proteasome	*PSMD3*	AC089998.1,AC105275.2,LINC01783
RNA transport	*RGPD1*	CSNK1G2.AS1,ITGB2.AS1,LINC01258,RP11.475I24.8
	*RPP30*	AC034229.3,AL391987.3,RMRP
Primary immunodeficiency, NF-kappa B signaling pathway	*TNFRSF13C*	AC110079.2,AL121603.2
Spliceosome	*USP39*	AC034229.3,AC092620.3,RMRP

Representative pathways identified from the KEGG database are indicated along with the differentially expressed mRNAs involved in each pathway. The third column of the table lists the lncRNAs correlated with the mRNAs and therefore potentially predicted to be involved in the corresponding pathways.

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
