# Peer review of "Profiling of RNAs from Human Islet-Derived Exosomes in a Model of Type 1 Diabetes"

_ijms, 2019, doi:10.3390/ijms20235903_

Round 1
Reviewer 1 Report
I would like to say the ms is well written. The originality and the content significance are high. I recommend to publish the ms in present form.
Author Response
I would like to say the ms is well written. The originality and the content significance are high. I recommend to publish the ms in present form.
Response: We thank the reviewer for his/her careful assessment of our manuscript and for the opportunity to publish our manuscript in the present form.
Reviewer 2 Report
The article is very interesstinf and carefully prepared.
It is based on modern methodology and current literature. However, I would like to clarify the issue of ethics committee approval here?
I believe that information on this subject should be included in the article.
I also have doubts about the application. I think the Researchers should add a bit more about the possible usefulness of the results.
Author Response
We thank the Reviewers for their careful assessment of our manuscript and for the opportunity to submit a revised version of our paper. We have addressed each of the Reviewer’s concerns in the point by point response below:
1. “The article is very interesstinf and carefully prepared. It is based on modern methodology and current literature. However, I would like to clarify the issue of ethics committee approval here?
I believe that information on this subject should be included in the article”.
Response: Thank you for highlighting this. Because we receive human islets that are de-identified, our study does not require a full IRB approval. However, our study was reviewed by the IRB and granted status as exempt from full board review. We have now included a statement on ethics committee approval in the manuscript.
2. I also have doubts about the application. I think the Researchers should add a bit more about the possible usefulness of the results.
Response: Thank you for your suggestion. We have now elaborated the last paragraph in the discussion to emphasize the usefulness of the results.
Previous studies have shown that the distribution of RNAs in exosomes is related to the parent cell type [46]. Since pancreatic islets are challenging to monitor for early disease development and progression, identification of molecular markers released from the islets may serve as indicators of a change in the cellular status. Circulating markers identified from patient derived samples are often challenged with the question of identifying the source of these markers because serum and plasma host a milieu of molecules from a variety of organs [47]. Studies such as ours may help narrow down the source of these circulating biomarkers and provide information about the trajectory of disease progression along with understanding the molecular functions of the identified biomarkers. However, the success of this approach will likely be dependent on the parallel ability to identify EV’s that are β cell-derived based on unique cell surface markers, as has been done in cancer (48). Quite often, there has been some controversy on the choice of endogenous reference controls for validation studies in human serum or plasma samples [49]. Comprehensive profiling of RNAs from islet-derived exosomes may give us an overview of the expression patterns of RNAs and provide a good base to search for endogenous controls in biomarker studies related to T1D. While many studies have focused on miRNAs extensively, our study includes the identification of newer classes of potential biomarkers for T1D as well such as piRNAs, snoRNAs, tRNAs and lncRNAs, whose roles as biomarkers have been demonstrated in other disease settings [16,22,50-54]. Validation of the study findings in clinical samples is warranted as it may open up opportunities to explore beyond the well-studied classes of RNAs, which can provide a basis for building a comprehensive and inter-molecular biomarker model for T1D.